# Postoperative Atrial Fibrillation: A Review

**DOI:** 10.3390/biomedicines12091968

**Published:** 2024-09-01

**Authors:** Sidra Shah, Vipanpreet Chahil, Ayman Battisha, Syed Haq, Dinesh K. Kalra

**Affiliations:** Division of Cardiology, Department of Medicine, University of Louisville, Louisville, KY 40202, USA; s0shah36@louisville.edu (S.S.); a0batt07@louisville.edu (A.B.); shhaq001@louisville.edu (S.H.)

**Keywords:** atrial fibrillation, perioperative, stroke, anticoagulation, warfarin, direct-acting anticoagulants, bleeding risk

## Abstract

Atrial fibrillation (AF) in the postoperative phase is a manifestation of numerous factors, including surgical stress, anesthetic effects, and underlying cardiovascular conditions. The resultant cardiac hyperactivity can induce new onset or exacerbate existing AF. A common phenomenon, postoperative atrial fibrillation (POAF) affects nearly 40% of patients and is associated with longer hospitalization stays, and increased mortality, heart failure, stroke, and healthcare costs. Areas of controversy in POAF include whether to anticoagulate patients who have short-lived POAF, especially given their higher bleeding risk in the postoperative period, and the identification of patients who would benefit the most from preventive drug therapy for POAF. This review discusses the pathophysiology and management of POAF, and strategies to reduce its occurrence.

## 1. Introduction

Atrial fibrillation (AF) is the most common arrhythmia, and there are approximately between 3 and 6 million people living with this condition in the US [1]. The incidence of AF increases with age; for example, individuals aged >40 have 2.3% compared to a 5.9% incidence in individuals 65 years and older [2]. AF is underdiagnosed, given that a third of the population is asymptomatic. There are numerous risk factors associated with the development of AF such as obesity, hypertension, a sedentary lifestyle, smoking, alcohol, sleep disorder breathing, thyroid disease, cardiovascular disease, chronic kidney disease, medications, and postoperative status. AF remains underdiagnosed and underestimated despite marked advancements in the diagnosis of AF. Post-operative atrial fibrillation (POAF) poses a financial burden on the healthcare system as it prolongs hospital length, results in higher utilization of healthcare resources, and is detrimental to the patient as it is associated with increased morbidity and mortality [3]. Patients who go on to develop POAF are four times more likely to develop stroke and twice more likely to die [4].

POAF is a significant complication after surgery in hospitalized patients. There is no definitive definition of POAF, as what constitutes POAF varies among authors and in the literature. Generalized definitions of POAF amongst various publications include any POAF that requires treatment, AF lasting greater than 30 s in a postoperative patient, new onset of AF post-operation regardless of the duration, or any AF post-operation lasting more than ten minutes [5].

POAF develops within the first six days of surgery in 90% of patients, correlating with the maximum intensity of a systemic inflammatory response [6]. Numerous mechanisms play a vital role in the development of POAF. An exact mechanism for POAF is not established; however, multiple etiologies are known to increase the risk of the development of POAF. Structurally, it has been theorized that atrial changes and pulmonary veins play a nidus of excitability, which increases the risk of developing POAF, as well as electrolyte disturbances leading to metabolic changes and altering electrical impulses [7]. Patient medical history plays a pivotal role in the development of POAF as conditions such as myocardial infarction, hypertension, heart failure, atrial fibrosis, heart disease, male sex, obesity, alcohol, sleep-disordered breathing, thyroid disease, and history of prior arrhythmias all have been linked to the onset of POAF [7].

Up until recent times, POAF was considered a self-limiting condition. However recent studies have shown that it is associated with increased ischemic stroke risk and mortality [2]. There are two major categories of POAF which can be divided into cardiac versus non-cardiac surgery. The incidence of POAF in non-cardiac surgeries is only 3% compared to 20–40% in cardiac and/or thoracic surgeries. It is noted that patients undergoing a coronary artery bypass graft (CABG) along with valvular surgery have the highest risk of developing POAF at 60–80%. Diagnosing and treating POAF is imperative as recent studies have shown similar long-term risk profiles of thromboembolic events between non-valvular AF and POAF [8]. The pathophysiology of POAF development in non-cardiac surgery is multifactorial, and factors such as activation of the sympathetic nervous system from the surge of catecholamines induced by stress from surgery, hypovolemia, electrolyte imbalance, and anemia are all vital components that play as a pathological precursor to the onset of POAF in non-cardiac surgery. Intraoperative fluid administration can result in hypervolemia resulting in stretching of the right atria which is a risk factor for the development of POAF. Therapies for POAF include anti-inflammatory drugs such as colchicine, statin, and BB. Devereaux et al. conducted a randomized control trial called POISE trial which consisted of 8000 patients undergoing non-cardiac surgery, and patients were randomized to either metoprolol succinate vs. placebo group. Patients in the metoprolol succinate group had 2.2% new-onset POAF compared with 2.9% in the placebo group (HR 0.76; 95% CI 0.58–0.99) [9]. The pathophysiology of POAF development in cardiac surgery involves similar factors as seen in non-cardiac surgeries however the pathological structural changes that occur with cardiac surgery increase the risk of development of POAF. Manipulation of the atrium during venous cannulation is a nidus of inflammation that results in heterogenicity of atrial conduction as seen in canine models. During cardioplegic arrest induced during CABG, there is suboptimal cooling of the atria which predisposes to POAF via reperfusion injury [10]. Anti-inflammatory medications play a pivotal role in preventing the onset of POAF. In the COPPS trial, Colchicine demonstrated a significant reduction in the onset of POAF compared to placebo (12% vs. 22%, relative risk reduction of 45%, and 95% CI 34.0–94.0) [10].

### Incidence

POAF is a common complication following cardiac surgery, affecting approximately 30% of patients overall. Incidence varies by procedure type: up to 50% after valve replacement, around 20% following CABG, and approximately 30% in aortic surgery patients. Combined valve and CABG surgeries exhibit even higher rates: up to 80%. In contrast, heart transplants have a lower POAF incidence of about 4% due to the denervation of the implanted heart. The incidence is also notable following transcatheter structural heart interventions, particularly transcatheter aortic valve replacement (TAVR), with in-hospital POAF rates of 4.3% versus 36.6% for TAVR and surgical aortic valve replacement, respectively. Thoracic surgeries see POAF rates between 10% and 20%, with higher rates in more invasive procedures like pneumonectomy, lung transplant, and esophagectomy (around 30%). Non-cardiac, non-thoracic surgeries have lower POAF incidences ranging from 0.4% to 15%, with colorectal surgery presenting the highest risk among these procedures. Differences in baseline patient risk profiles, surgical intervention details, and postoperative monitoring contribute to the variability in POAF incidence rates [5,11,12,13,14,15,16,17].

A prospective, randomized study by Ascione et al. involving 200 patients undergoing CABG, either with cardiopulmonary bypass (CPB) and cardioplegic arrest or off-pump surgery, showed CPB as a critical independent predictor of POAF. The study found that patients who underwent on-pump surgery had a markedly higher incidence of sustained AF (39%) compared to those who had off-pump surgery (8%). The research highlighted that CPB, inclusive of cardioplegic arrest, significantly elevates the risk of POAF, with an odds ratio of 7.4, suggesting that the myocardial ischemia and inflammatory responses associated with CPB play a pivotal role in the development of this arrhythmia [18].

A prospective multicenter study of 4657 patients undergoing surgery reported that most of the first episodes of POAF occurred by day two. In contrast, the majority of recurrent episodes occurred by day three. More than 40% of patients with POAF had more than one episode [19]. Abdelmoneim et al. highlighted the recurrent nature of POAF beyond the initial hospitalization. Within the first month post-discharge, 80% of patients had their first recurrence, and 76% experienced at least one recurrence during the first year. Beyond one year, 30% of patients continued to have recurrences. The persistent recurrence suggests that POAF is not a transient postoperative phenomenon but may represent a more chronic condition requiring long-term management strategies [20]. The recurrence of POAF is associated with older age, left ventricular hypertrophy, aortic atherosclerosis, and specific surgical techniques such as bicaval venous cannulation. Furthermore, the withdrawal of beta-blockers (BB) or angiotensin-converting enzyme inhibitor therapy postoperatively significantly increases the risk of recurrent POAF. At the same time, the use of these medications, along with potassium supplementation and nonsteroidal anti-inflammatory drugs (NSAIDs), has been shown to reduce the incidence [19]. Despite many cases resolving within 24 h, the persistent and recurrent nature of POAF not only prolongs hospital stays but also increases the risk of complications such as cognitive changes, renal dysfunction, and infections, highlighting the need for effective management strategies to mitigate its impact on patient outcomes [6].

## 2. Risk Factors

POAF is related to a broad spectrum of risk factors, including patient-related, surgery-related, and reversible risk factors. A patient’s medical problems predispose them to develop POAF, and such conditions can be modifiable and non-modifiable. Patient-related risk factors for POAF include age, male sex, hypertension, congestive heart failure, chronic obstructive pulmonary disease, ischemic heart disease, pre-existing AF, stroke, obstructive sleep apnea (OSA), and diabetes mellitus. Along with postoperative-related changes such as structural alteration, electrolyte disturbances, and inflammation-induced pathological fibrosis, all of which increase the risk of POAF (Figure 1).

Hypertension increases the risk of developing POAF by 29%, diabetes increases the risk of POAF by 6%, chronic obstructive pulmonary disease by 36%, heart failure by 56%, and myocardial infarction by 18% [6].

Advanced age is an essential factor associated with the development of POAF. Aging results in a pathological decrease in myocardial fibers, which are pivotal for elasticity. Aging is also positively associated with fibrinogenesis and collagen deposition in the atria. These atrial changes result in conduction abnormalities, especially when the pathological changes involve the sinoatrial node. A large single-institution retrospective study involving 14,960 patients spanning over two decades who underwent cardiac surgery demonstrated a non-linear trend between age and POAF incidence, showed a higher incidence of POAF past age 55 and a five-fold increase in developing POAF aged 72 or older compared to patients aged less than 55 [21].

Cardiac and abdominal surgeries carry a higher risk of developing POAF compared to other surgeries. There are roughly between 40 and 50 million non-cardiac surgeries performed in the US annually, of which one-third of the patients are 65 years and older. The patients who had episodes of hypotension lasting longer than 10 min intraoperatively had a significantly increased risk of POAF given hypotension-induced hypoxemia leading to cardiac myocyte pre-excitability [2]. Cardiac surgery results in atrial structural changes through inflammation and structural changes via surgery, as these changes are more likely to result in POAF. Atrial changes secondary to cardiac surgery can be detected on electrocardiogram (EKG) as they cause conduction variation in the atria which can be demonstrated by a postoperative EKG showing p-wave changes compared to preoperative EKG. P-wave duration and dispersion, defined as the difference between the duration of the longest and shortest p-wave, were studied in 300 patients who underwent cardiac surgery, and postoperative p-wave dispersion was significantly shorter in the postoperative patients [22]. Lastly, reversible causes of POAF need to be addressed aggressively before the operation to minimize any complications that can increase morbidity and mortality. Reversible factors that should be addressed before surgery include electrolyte derangements, hypoxia, infection, anemia, hypovolemia/hypotension, hypothermia, alcohol or benzo withdrawals, hyperthyroidism, and holding home medications such as BB and calcium channel blockers, which can lead to increased sympathetic tone leading to tachycardia [2].

OSA is an independent factor associated with the development of POAF. OSA is the intermittent partial or complete collapse of the upper airways during sleep which promotes episodes of hypoxemia, hypercapnia, increased sympathetic tone, intermittent arousals, and changes in intrathoracic pressure which result in increased inflammatory activity, endothelial dysfunction, and remodeling of the atria [23]. All of the above-combined pathological changes result in an increased risk of POAF. More importantly, it was observed that the increased severity of the OSA correlated with an increased likelihood of developing POAF independent of other factors. RICCADSA, a cohort study of 147 adults who underwent revascularization in Sweden between September 2005 and November 2010, showed that 32% of the total patients developed POAF, including a strong correlation between the severity of the OSA and the risk of developing POAF. The severity of OSA was measured using the apnea–hypopnea index (AHI). The incidence of POAF was 11.1% for (AHI < 5), 29.2% for (AHI 5–14), 30.4% for (AHI 15–29), and 44.9% for severe OSA with (AHI 30 or higher) post-operatively [23].

## 3. Prediction

There are multiple scoring systems used to predict postoperative atrial fibrillation. POAF, CHA2DS2-VASc, and HTACH scores have been shown to have good predictive value. An accurate scoring system is vital to increasing the positive predictive value of a scoring model to predict patients at higher risk for developing POAF. This will help lower the burden on healthcare systems and decrease the utilization of healthcare resources. A single cohort using retrospective analysis of prospectively collected data between 2010 and 2016 of 3113 patients undergoing cardiac surgery was conducted to determine the predictability of POAF. Discrimination and calibration were the two components studied between the studies to assess the predictability. CHA2DS2-VASc demonstrated the highest predictive ability [24].

## 4. Pathophysiology

POAF pathophysiology is multifactorial and influenced by structural and cellular changes occurring post-operation. Structural changes predisposing to POAF include atrial structural changes, pericardial effusion and inflammation, peri-atrial adipose tissue metabolic activity, and myocardial ischemia (Figure 2). In contrast, cellular-level alterations include gap junction uncoupling, ion channel modifications, and re-entry/ectopic activity in the pulmonary veins [5]. During CABG, there is a direct manipulation of the atria, such as venous cannulation, which occurs via right atriotomy, and, in mitral or tricuspid valve surgery, there is perivalvular atrial tissue that is put under mechanical stress resulting in conduction changes [5]. This pathological state gives way to the enhancement of pathological arrhythmias, including POAF. Postoperative inflammation activates fibroblasts, which produce cytokines and chemokines, causing the fibroblasts to replace epicardial and myocardial myocytes with fibrosis by the fibroblasts in response to inflammation. Connexin 40 and 43 are proteins highly expressed in cardiac myocytes, including in the atria. Inflammation leading to fibrosis replaces the physiological myocytes with pathological fibrosis. This change increases the risk of developing POAF as decreased expression of Connexin 40 and 43 leads to slow and non-uniform conduction through the atria, which is arrhythmogenic [5].

Inflammation is another major factor that promotes POAF. Inflammatory markers include CRP peak at 48 h post-surgery, which correlates with the timeframe of the highest risk of developing POAF. A study by Melby et al. conducted a time series analysis of patients undergoing CABG and valvular surgeries and demonstrated two periods, immediately postoperative and 48 h post-surgery, with the highest likelihood of developing POAF (see ref. [10]). Peri-atrial adipose tissue metabolic activity has been linked to the development of POAF. Pro-inflammatory markers released by peri-atrial adipose tissue cause decreased conduction speed and variability of conduction in cardiomyocytes, which all increase the growth of AF. It is noted that cardiac adipose-mediated interleukin 1β (IL-1β) secretion is much higher in patients who go on to develop POAF compared to patients with similar surgeries who do not develop POAF [5]. This observation sheds light on the role inflammation plays in the onset of POAF and how some individuals are at increased risk of developing POAF compared to others with similar surgeries.

Cardiac surgery results in significant physiological changes, including vasoplegia, excessive catecholamine release, large fluid shifts, and neurohormonal changes, which all predispose to POAF [10]. Postoperative pain increases sympathetic tone, which leads to tachycardia and arrhythmogenesis. Sympathetic overtone increases calcium entry and release from the sarcoplasmic reticulum, leading to hyperactivity of ryanodine receptors and inducing calcium influx, which results in arrhythmias [25,26]. Sympathetic overtone not only causes increased heart rate and contractile force but also results in upregulation in excitability and automaticity. This phenomenon is further supported by the observation that an increase in heart rate and atrial ectopic activity precedes the onset of POAF. A study by Melo et al., which consisted of 207 low-risk CABG patients who received ventral denervation, demonstrated a decreased incidence and severity of POAF (see ref. [27]). Fluid shifts also contribute to the onset of POAF. Patients are administered various anesthetic medications intraoperatively. Intraoperative blood loss, which can cause hypotension, along with other various preop conditions, such as dehydration, decreased oral intake, and anti-hypertensive medications, can all increase the risk of hypotension requiring extensive volume intravenous fluid administration, which can cause significant intravenous fluid shifts leading to right atrial stretching which directly increases the risk of developing POAF [8]. Atrial stretching leads to refractoriness and delayed conduction, which can perpetuate multiple reentry pathways, further increasing the risk of POAF development [3].

Electrolyte derangements are common amongst patients undergoing cardiac surgery as patients are on an insulin drip for tight glycemic control preoperatively as well as multiple other medications which can all cause electrolyte disturbances. Magnesium has been studied to play a significant role in the risk of developing POAF. TRPM7 channels play a pivotal role in activating fibroblasts to myofibroblasts which causes fibrosis to replace the physiological cardiac myocytes, and the TRPM7 channels are activated by low magnesium levels. The Framingham Heart Study showed that hypomagnesemia post-cardiac surgery was an indicator of developing POAF [6]. High levels of magnesium increase atrioventricular node conduction time, which protects against oxidative damage. Hypomagnesemia increases atrioventricular node automaticity [21].

POAF development in OSA patients involves multiple factors that increase the likelihood of developing POAF. Numerous mechanisms predispose to the development of POAF, such as the imbalance between the sympathetic and parasympathetic systems, negative thoracic pressure during apnea, and chronic intermittent hypoxia, which all play a role in changing the atria chemically, or structurally. During apnea episodes, there is increased tone of the parasympathetic system as noted by prominent bradycardia which is followed by rebound tachycardia, likely from the withdrawal of the parasympathetic system which results in increased tone of the sympathetic system, and it manifests as increased blood pressure. Autonomic dysfunction has been linked to the development of atrial structural remodeling which makes the atria vulnerable to re-entry thus leading to the development of POAF [28]. On the other hand, negative thoracic pressure (NTP) occurs during apnea when respiratory muscles generate a breath against an occluded airway. NTP in the chronic setting leads to atrial stretching, which results in atrial fibrosis and increased ectopic activity. NTP also results in baroreceptor activation, which causes increased parasympathetic overtone resulting in electrical remodeling and dysfunction [28]. Lastly, chronic intermittent hypoxia causes increased levels of hypoxia-inducible factor (HIF)-1α and promotes upregulation of nuclear factor-κB which leads to diverse inflammatory responses [28]. Inflammation results in pathological fibrosis, necrosis of the physiological tissue, and dysregulation of gap junction proteins, such as connexin-43, leading to slowing of the conduction.

## 5. Treatment

Treatment and management of POAF is a holistic process that, first and foremost, involves achieving hemodynamic stability. In conjunction with this, provoking casualties should be identified and promptly addressed. As alluded to previously, POAF is usually an insidious manifestation of underlying cardiovascular disease revealed due to postoperative stress. The patient may have hypertension, coronary artery disease, valvulopathy, or heart failure that is undiagnosed or has progressed. Therefore, the vast array of therapies is carefully selected to complement treatment for underlying etiologies while avoiding further deleterious effects (Figure 3 and Figure 4).

Beta-blockers are the most commonly reviewed intervention for the prevention of POAF. They are a Class 1 recommendation per the 2020 European Society of Cardiology [ESC] guidelines for POAF after cardiac surgery and have been related to a reduced incidence of POAF. This benefit is due to the attenuation of the sympathetic tone which directly relates to the arrhythmia initiation and atrial refractoriness [29]. It is recommended to continue the BB throughout the entire perioperative period for about four to six weeks to prevent withdrawal. The most common BB used in the trials was propanol. However, in non-cardiac surgery, BB initiation perioperatively is not recommended per the international guidelines due to an increased risk of death and stroke seen in a randomized controlled trial with extended-release metoprolol [30]. In the older literature, sotalol was compared to commonly used BB. It was found that higher doses of sotalol had greater side effects, such as pro-arrhythmic effects. Therefore, sotalol is a class IIb recommendation in POAF prevention [31].

Calcium channel blockers are considered a Class 1 indication to achieve rate control when a BB cannot be used in patients due to intolerance or contraindications. Non-dihydropyridine calcium channel blockers, such as diltiazem and verapamil, are classified as Class IV antiarrhythmic drugs for the prevention of POAF in cardiac and non-cardiac surgery. This class of medications has been seen to reduce supraventricular arrhythmias specifically after cardiac surgery. This was seen in a meta-analysis study consisting of 41 studies, which had a total of 3327 patients, which showed that non-dihydropyridine calcium channel blockers did reduce the incidence of myocardial infarction, ischemia, and supraventricular tachycardia (95% CI from 0.41 to 0.93) including atrial flutter and AF [32]. However, the use of calcium channel blockers is limited as they can be associated with low output syndrome and increased atrioventricular blocks [32].

Amiodarone can also be used for the acute management of POAF in elevated-risk patients as it is seen to suppress ectopic triggers and the development of ventricular arrhythmias. It is a class III antiarrhythmic drug that blocks potassium and calcium channels. Based on the 2020 ESC guidelines, perioperative amiodarone use for POAF is a class IA recommendation [33]. However, it is less commonly used due to requiring loading doses in the days before surgery and its side effect profile. Preoperative amiodarone has a Class IIa recommendation (ACC/AHA/ESC) for POAF prevention after cardiac surgery [34]. While amiodarone does successfully maintain sinus rhythm, it does come with many side effects to be aware of, including bradycardia, hypotension, pulmonary toxicity, and liver and thyroid dysfunction. Studies have shown similar efficacy of BB and amiodarone [31,34]. A large study called PAPABEAR (Prophylactic Amiodarone for the Prevention of Arrhythmias that Begin Early After Revascularization, Valve Replacement, or Repair) trial showed that the use of oral amiodarone resulted in a large reduction in the incidence of postoperative atrial tachyarrhythmias.

NSAIDs have also been seen to decrease the risk of POAF. The NAFARM Randomized Trial studied the effect of naproxen on the prevention of AF after CABG. It was concluded that postoperative naproxen use was found to decrease the duration of AF; however, it did not reduce the incidence of AF [35]. Colchicine is an inexpensive anti-inflammatory drug that can be used in treating POAF. The COP-AF trial hypothesized that colchicine minimizes the incidence of POAF in patients undergoing non-cardiac thoracic surgery. The randomized trial consisted of participants receiving oral colchicine 0.5 mg twice daily or receiving placebos. The participants started this four hours before surgery for ten days. The study included 3209 patients aged 55 years and older who were having major non-cardiac thoracic surgery. The results showed no significant difference between the colchicine treatment group versus the placebo group. It was deemed that colchicine did not significantly reduce the incidence of POAF [36]. In contrast, the COPPS trial showed colchicine, in comparison to a placebo, reduced the incidence of POAF and caused faster conversion to sinus rhythm (12% vs. 22%) [32]. Therefore, post-op colchicine is a Class IIb recommended treatment in European guidelines [14].

Due to hypomagnesemia, which is a frequent occurrence after cardiac surgery, magnesium has commonly been used in POAF for its pro-arrhythmia effect. It is shown to reduce the incidence of POAF after cardiac surgery. However, the full reason for the efficacy of intravenous magnesium is uncertain as there has been conflicting data [37]. For example, in a randomized control trial with 389 patients undergoing cardiac surgery, a group of patients received magnesium in a 50 mg/kg bolus after anesthesia induction, followed by an additional 50 mg/kg infusion over 3 h, and this was found to have no significant impact in terms of reducing the incidence of POAF versus the placebo group. Another meta-analysis study of five high-quality randomized control trials reported no meaningful reduction in POAF (95% CI 0.61–1.44) [38].

### Prevention

The use of fish oil, rich in omega-3 fatty acids, has been studied for its potential benefits in various cardiovascular conditions, including POAF. The Omege-3 Fatty Acids for Prevention of Post-Operative Atrial Fibrillation (OPERA) randomized trial was a double-blind, randomized clinical trial completed with 1516 cardiac surgery patients. The patients were randomized to receive fish oil (1 g capsules) or a placebo. The dosing included a preoperative loading of 10 g over three to five days. Then, postoperatively, 2 g/day until the patient was discharged or postoperative day 10, whichever was first. The study measured POAF episodes lasting longer than 30 s. Overall, the study showed that fish oil, compared to placebo, did not reduce the risk of POAF [5]. Therefore, the use of fish oil in POAF remains unclear.

There is limited data on the use of botulinum toxin injection for the prevention of POAF. It has mainly been studied in animal models. However, there was a small randomized control trial consisting of 60 CABG patients who had a history of paroxysmal atrial fibrillation. The incidence of POAF was 7% in the intervention group versus 30% in the control group [5].

Specific research on different types of anesthesia used in surgery has also been studied for POAF prevention. A study by Liu and colleagues looked into the use of propofol versus dexmedetomidine for sedation. It was found that there was a lower incidence of POAF with the use of dexmedetomidine, which was statistically significant (13.6% in dexmedetomidine versus 36.4% in propofol use). This preventative effect is likely due to lower sympathetic tone, reduced inflammatory response, and increased refractory period by altering calcium currents [39].

These interventions can help patients lower the incidence of POAF, decrease the length of hospital stay, and reduce overall costs. Figure 3 provides further details of the common drug interventions in POAF.

## 6. Management of Anticoagulation

There are not many studies in the current literature that elaborate on the role of short-term oral anticoagulation in patients with POAF, whether in cardiac or non-cardiac procedures with regards to stroke prevention. Often the decision to anticoagulant is subject to the type of surgery, bleeding risk versus thromboembolic risks—including prevalence of known prior deep vein thrombosis or pulmonary embolisms—and, finally, individualized preference. The decision-making process for anticoagulation in POAF requires a careful review between reducing the risk of stroke and managing the potential for bleeding complications. This process typically involves evaluating the bleeding risk by reviewing the patient’s history for prior bleeding events, including gastrointestinal bleeding or hemorrhagic stroke, and other factors including age and comorbidities. Risk assessment tools such as CHA2DS2-VASc score and HAS-BLED can help a provider make a decision about anticoagulation. The Lin et al. meta-analysis that reviewed 35 studies with 2,458,010 total participants showed that POAF did correlate with an increased risk of short- and long-term adverse effects, including stroke and overall mortality. A total of 62% demonstrated a greater risk of early stroke events and a 44% risk of mortality compared to patients without POAF [40]. This study also recommended starting anticoagulation in women with a CHA2DS2-VASc score greater than 2 [40]. Per the 2020 European Society of Cardiology (ESF) AF guidelines, long-term oral anticoagulation can be considered in patients with POAF after cardiac surgery who are at risk for stroke; this is a Class IIb recommendation. The 2020 CCS guidelines recommended following patients with POAF after cardiac surgery indefinitely as they demonstrate a propensity for recurrence of AF. However, further elaboration and specific recommendations are limited [4]. The 2023 ACC/AHA/ACCP/HRS guideline on AF recommended that patients with greater risk for strokes or thromboembolic events be started on anticoagulation based on risk scores [41].

Once the need for anticoagulation has been deemed appropriate, contingent on timeline and risk stratification scores, initiation of therapy should be pursued. Typically, in POAF lasting longer than 48 h and those with a CHADS2 score greater than 2, anticoagulation is required. For a single episode of AF lasting less than 48 h, anticoagulation typically is not recommended unless high-risk factors exist, such as mitral stenosis. Short-lived POAF situations often present as a clinical dilemma due to the lack of definitive evidence supporting the benefit of adding anticoagulation therapy. The transient nature of POAF episodes, which may resolve spontaneously in a brief period, raises questions about the overall risk of stroke and the potential advantages of anticoagulation in these cases. Therefore, there is variability in clinical practice and a need for further research to establish more definitive guidelines. It is important to recognize patients with secondary AF have a higher predisposition for recurrent events. The Framingham Heart Study reported a 42% 5-year recurrence, 6% 10-year incidence, and a 62% 15-year incidence in new AF with secondary precipitant [42]. Anticoagulation should be continued for four weeks after surgery, followed by reassessment to determine the need for continuation. To assist with this, a cardiac event monitor can be considered post-operatively at discharge to further assess the recurrence or persistence of AF. If multiple episodes of recurrences are noted following the four weeks, anticoagulation should be continued long-term (Figure 5) [37].

Direct-acting oral anticoagulants (DOACs) are currently first-line therapy for all patients with AF except for those with mechanical valves or moderate–severe mitral stenosis. These DOACs include apixaban, dabigatran, edoxaban, and rivaroxaban. Multiple studies have shown DOACs to have a lower bleeding rate and risk for intracranial hemorrhage when compared to warfarin. Adjustments should be made based on age, renal function, and lower body weight [30]. On the contrary, DOACs provide the convenience of structured dosing, lack of dietary restrictions, and fewer drug interactions. As a result, DOACs have gained increased notoriety and carry a more favorable profile amongst patients and practitioners. It is, however, important to recognize that a major drawback to using DOACs remains the cost and concerns for reversal during active bleeding.

## 7. Periprocedural Interruption and Postprocedural Bridging

Many factors can dictate the need for bridging. Depending on the type of anticoagulation and intended procedure, anticoagulation cessation may be anywhere from 2 to 5 days before maintaining appropriate intraprocedural hemostasis. Recommendations are similar regarding when to hold and resume anticoagulation after both cardiac and non-cardiac surgery. Per CHEST guidelines, patients on vitamin K antagonists such as warfarin should be discontinued at least five days before surgery. Following surgery, warfarin can be resumed within 24 h if adequate hemostasis and bleeding control are present. However, it is encouraged to bridge anticoagulation in patients with a high risk for thromboembolism and/or who have a mechanical heart valve. Anticoagulation can be continued without bridging for minimally invasive procedures, such as dental, ophthalmology, dermatology, colonoscopy, or pacemaker placement. In contrast, DOAC therapy is recommended to be stopped before surgery depending on the type of surgery and associated risks of bleeding. DOACs can be discontinued one to four days before the procedure and resumed 24 to 72 h after.

During the preprocedural washout period, low molecular weight heparin (LMWH) or unfractionated heparin (UFH) can be used for maintenance or “bridging”. The heparin bridging regimen consists of providing a therapeutic dose of LMWH such as enoxaparin 1 mg/kg twice daily or 1.5 mg/kg daily or a full dose of UFH [40]. These recommendations decrease the probability of adverse events, including thromboembolism or stroke, and should be distinguished from heparin bridging.

Findings from the Outcomes Registry for Better Informe Treatment of Atrial Fibrillation (ORBIT-AF) in a study by Steinberg et al. explored the outcomes of temporarily stopping anticoagulation and following patients who underwent bridging with LMWH or UFH versus patients who were not bridged. The ORBIT-AF is a national registry of 10,312 patients with AF. Inclusion major criteria included age greater than 18, EKG showing AF not attributed to a reversible cause, and a maximum 3-year follow-up. Only interruptions for procedures were recorded, and the data included both cardiac and noncardiac procedures, bridging anticoagulant used (LMWH, UFH, fondaparinux, or other), and adverse events associated with interruption. These events included bleeding or thrombotic events. In the study, 665 interruption events were noted that involved bridging anticoagulation. It was seen that LMWH was used in 73%, UFH in 15%, fondaparinux in 1.1% of patients, and another anticoagulant in 11%. Reportedly, 2138 patients did not undergo bridging. Those who underwent bridging were more likely to have had valvular heart disease, mechanical prosthetic valves, heart failure, and a previous history of strokes. Therefore, the mean CHA2DS2-VASc was higher in bridged patients (4.25 vs. 4.03). The study concluded three things: (1) OAC interruptions were common in AF patients undergoing cardiac or noncardiac surgery, even if it entailed a minimally invasive procedure, (2) bridging anticoagulation was utilized in one-fourth of the patients, and the choice to bridge was determined based on the patient’s bleeding risk, and (3) bridged patients during periprocedural period did not have reduced complications, in fact, it was associated with a higher rate of bleeding, thrombotic, and adverse events [43].

Other studies propose contradicting findings. The RE-LY study compared warfarin to dabigatran and demonstrated that patients who received bridged therapy with warfarin had more thromboembolic events versus those who did not receive the treatment (1.8% vs. 0.3%). There was also a higher risk of bleeding in patients with warfarin and bridging. It went on to further suggest bridging is not needed with DOACs [44]. In the BRIDGE trial (bridging anticoagulation in patients who require temporary interruption of warfarin therapy for elective surgery), 1722 were divided into bridging and non-bridging groups. The study concluded that bridging with LMWH was non-inferior at preventing arterial thromboembolism but was associated with an increased bleeding risk (*p* = 0.0005) [45]. It is essential to consider that this trial did present a lower mean CHADS score of 2.3, with only over a third of participants having a score greater than 3. Furthermore, the trial did not represent major surgical procedures typically thought to be associated with high rates of thromboembolism and bleeding, such as cardiac or neurosurgery, carotid endarterectomy, or malignancy-related surgeries.

Regardless, initiating early anticoagulation is a Class IIa recommendation (level of evidence B) per international and US guidelines in preventing thromboembolism [6]. However, given the accumulating data, a re-evaluation of bridging criteria may be warranted as more harm can be appreciated than benefit. Patients on DOAC therapy would not benefit from bridging, and, in fact, are predisposed to a greater bleeding risk. Some literature proposes that bridging be reserved for high thromboembolic risk patients on warfarin. These risk factors would include a prior embolic event with anticoagulation cessation or on therapeutic anticoagulation; an episode of a transient ischemic attack or a cerebrovascular accident within the past three months; the presence of a mural thrombus or clot within the left atrial appendage; a mechanical mitral valve; venous thromboembolism within the past three months; a diagnosed hypercoagulable state [44].

## 8. Future Directions

Artificial Intelligence (AI) is increasingly recognized as a transformative tool in managing POAF. Clinicians can analyze large volumes of patient data to uncover subtle patterns and identify key risk factors predictive of POAF. Machine learning (ML) algorithms can predict the likelihood of POAF with high precision, as shown in Table 1, enabling healthcare providers to implement preventive measures and tailor postoperative treatment to individual patient profiles. This approach not only improves clinical outcomes by reducing the incidence and severity of POAF but also enhances resource utilization and operational efficiency within healthcare systems.

AI/ML models have been used to improve AF patient management, including drug dosing, procedural success, and treatment outcomes. A study by Levy and colleagues evaluated the use of reinforcement learning for optimizing dofetilide dose adjustments during initiation in 354 patients. The study found that dose adjustments, especially at later stages, reduced the likelihood of successful initiation, with a reinforcement learning algorithm predicting dosing decisions with 96.1% accuracy. Successful initiation was achieved in 87.1% of patients, potentially reducing healthcare costs by identifying patients unlikely to benefit from prolonged initiation attempts [59].

Vinter’s research involved 332 female and 790 male patients with persistent AF undergoing electrical cardioversion, achieving success rates of 44.9% in women and 49.9% in men. The best accuracy results for predicting successful cardioversion were 60% for women using logistic regression and 59% for men using ML models [60]. Alhusseini et al. involved 35 patients with persistent atrial fibrillation, utilizing 175,000 image grids for analysis. The convolutional neural network achieved a 95.0% accuracy in classifying AF patterns, outperforming traditional methods like support vector machines and linear discriminant analysis [61]. Ghrissi et al. included 16 patients with persistent AF, analyzing 23,082 multichannel electrogram recordings. The ML model achieved a sensitivity increase from 50% to 80%, maintaining accuracy and AUC around 90% with data augmentation techniques [62].

Luongo and colleagues explored the use of ML to predict atrial fibrillation driver locations and the success of pulmonary vein isolation (PVI) using 12-lead ECG data. Involving 46 patients (23 with PV-dependent AF and 23 with extra-PV sources), the classifier achieved 82.6% specificity and 73.9% sensitivity, showing a high potential for noninvasive AF driver localization [63].

Future research should focus on enhancing ML models for predicting POAF by integrating more data from electronic health records and utilizing real-time monitoring through wearable or implantable devices. This integration can improve the accuracy of POAF prediction and facilitate prompt management strategies. Additionally, ML models should aim to personalize therapeutic interventions, such as antiarrhythmic and anticoagulant therapies, and predict POAF-related complications like stroke and mortality. Real-time monitoring systems and wearable technologies can provide AI-driven early POAF detection, facilitating timely intervention. For instance, continuous ECG monitoring with CardioSTAT patches improved 30-day POAF detection by 17% in a pilot trial. However, integrating AI into clinical practice requires rigorous validation and prospective studies to ensure its efficacy and safety within the context of POAF management [64,65,66].

## 9. Conclusions

POAF is the most common type of arrhythmia in surgical patients and is associated with increased morbidity, mortality, and increased healthcare resource utilization. Alterations in cellular and ion channel function, inflammatory pathways, and sympathetic signaling along with comorbidities play a pivotal role in the development of POAF. The incidence of POAF is higher in cardiac surgeries (20–40%) compared to non-cardiac surgeries (3%). Post-operative inflammation results in fibrosis, gap junction uncoupling, the release of cytokines resulting in oxidative stress, and structural atrial changes occurring from direct alteration of the atria during surgery, all of which predispose to the development of POAF. Comorbidities such as COPD, hypertension, diabetes mellitus, congestive heart failure, OSA, smoking, obesity, advanced age, and being male all increase the risk of developing POAF.

The management involves a multifaceted approach, including both pharmacological and non-pharmacological interventions. BB and amiodarone are commonly used for rate and rhythm control. Anti-inflammatory drugs such as NSAIDs and colchicine have been shown to decrease the incidence of PAOF by decreasing the inflammatory response thus preventing fibrosis and conduction abnormalities which predispose to reentry pathways. It is essential to address comorbidities that pose a severe risk factor preoperatively along with correction of electrolytes, hypoxia, and maintaining euvolemia to minimize risk for POAF.

Anticoagulation may be considered based on an individual’s risk factors. For patients already on anticoagulation, it is essential to use one’s clinical judgment along with guideline recommendations to make the best decision for the patient. POAF represents a significant clinical challenge with potential implications for patient outcomes. The intricate interplay of surgical stress, inflammation, and cardiovascular health factors underscores the need for a comprehensive approach to prevention and management. Preoperative risk assessment, optimization of underlying cardiovascular conditions, and careful medication management are crucial components in mitigating the risk of POAF.

## Figures and Tables

**Figure 1 biomedicines-12-01968-f001:**
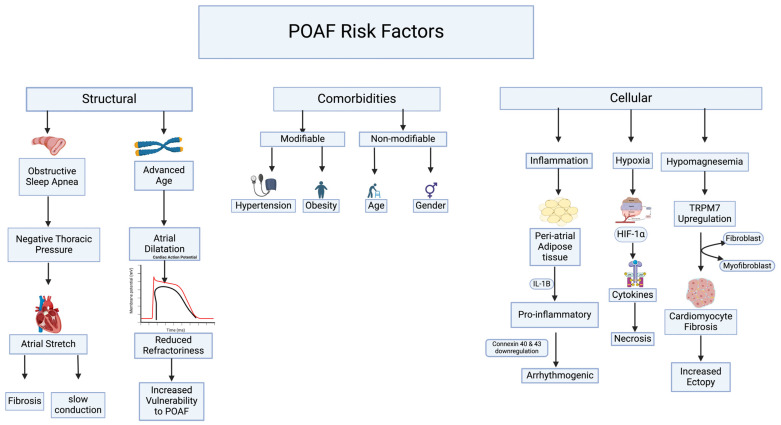
Risk factors of postoperative atrial fibrillation. These risk factors are important to address before surgery to minimize the incidence of POAF. Structural changes including atrial stretching by long-standing OSA result in the replacement of cardiomyocytes with pathological fibrosis which predisposes to conduction variation, and increased vulnerability to reentry pathway. Advanced age is the biggest independent risk factor for the development of POAF through atrial structural changes such as atrial dilatation consequently resulting in reduced refractoriness. Patient comorbidities, inIcluding modifiable ones, such as hypertension and obesity, and non-modifiable comorbidities, like advanced age and male sex, all play a pivotal part in chronic inflammation and remodeling which increase the risk of developing POAF. Changes at the cellular level through various mechanisms all aid in increasing the risk of POAF. Post-surgical inflammation which peaks at 48 h results in cytokines such as IL-1β resulting in an immense inflammatory response leading to the downregulation of gap junction proteins such as Connexin 40 and 43 which are essential in the propagation of action potentials across cardiac myocytes, causing arrhythmogenic state. Chronic intermittent hypoxia as seen in OSA patients causes increased expression of hypoxia-inducible factor-1α (HIF). HIF upregulates the expression nuclear factor-κB causing remodeling, and pathological changes. Hypomagnesemia is one of the major electrolytes known to increase atrial ectopy by promoting fibrosis through the activation of fibroblasts into myofibroblasts by TRPM7 expression.

**Figure 2 biomedicines-12-01968-f002:**
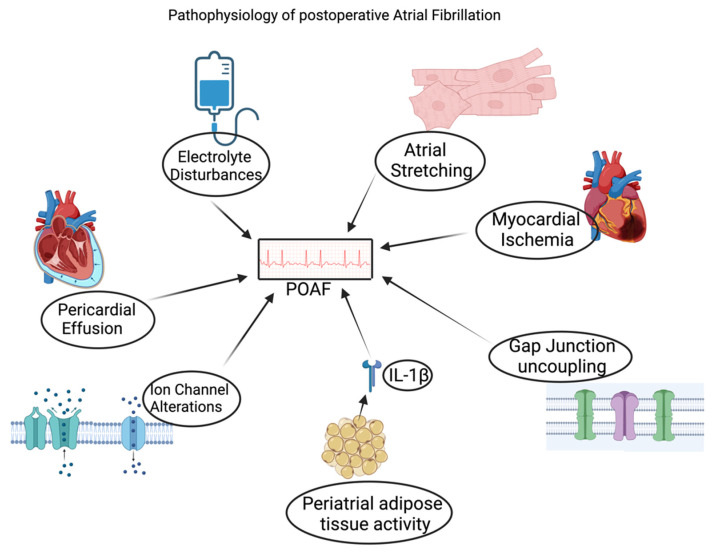
The pathophysiology of POAF is multifactorial and is influenced by modifiable and non-modifiable risk factors. Electrolyte disturbances, including hypomagnesemia, have been shown to activate TRPM7 channels responsible for activating fibroblasts to myofibroblast, resulting in fibrosis of the physiological cardiac myocytes, increasing the risk of POAF. Pericardial effusion is a common occurrence after coronary artery bypass graft and cytokine-mediated inflammation leads to the development of effusion, which increases the risk of POAF. Post-operation inflammation is a physiological response including released cytokines that mediate the inflammatory process. IL-1β is a major pro-inflammatory marker that results in inflammation of the cardiomyocyte which predisposes to slowing of cardiac conduction along with variation in the conduction and it is this mechanism that increases the chance of POAF. Connexin 40 and 43 are gap junction proteins expressed in the atria that are downregulated in inflammation, leading to decreased conduction velocity and arrhythmogenesis. Myocardial ischemia can result in atrial infarction, and ischemia is arrhythmogenic. Atrial stretching is another pathological change that occurs through various mechanisms, such as fluid shifts, as seen in hypervolemia, direct structural changes noted post-surgery, and long-standing comorbidities including hypertension, chronic obstructive pulmonary disease, and obesity.

**Figure 3 biomedicines-12-01968-f003:**
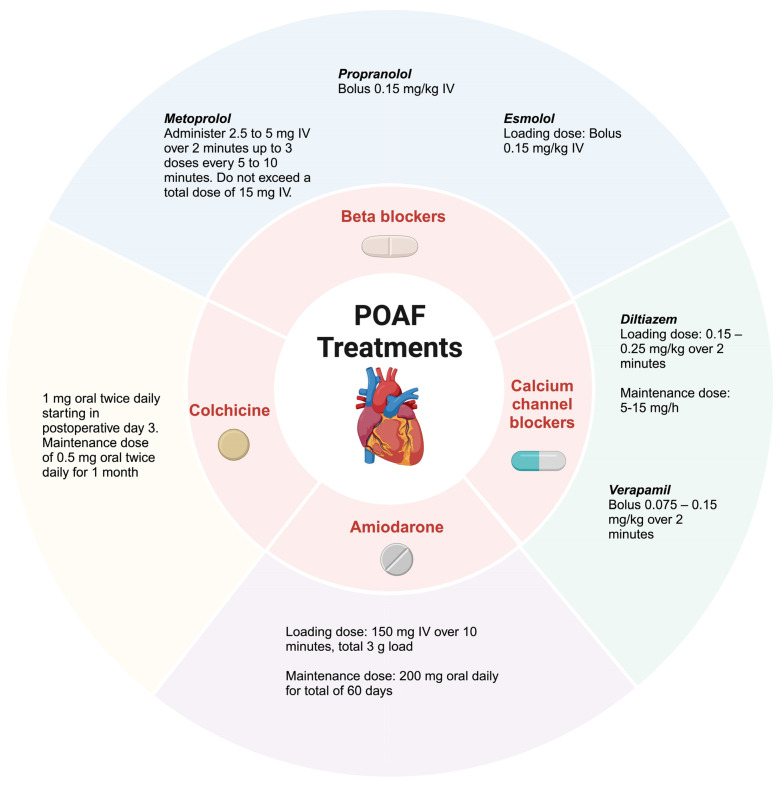
This figure illustrates the spectrum of medications commonly used to manage POAF. They are mainly categorized by rate control agents, rhythm control agents, and NSAIDs. Each class of medication addresses a different aspect of POAF management. The choice of medication depends on individual patient factors, including the severity of the arrhythmia, underlying comorbidities, and risk of adverse effects.

**Figure 4 biomedicines-12-01968-f004:**
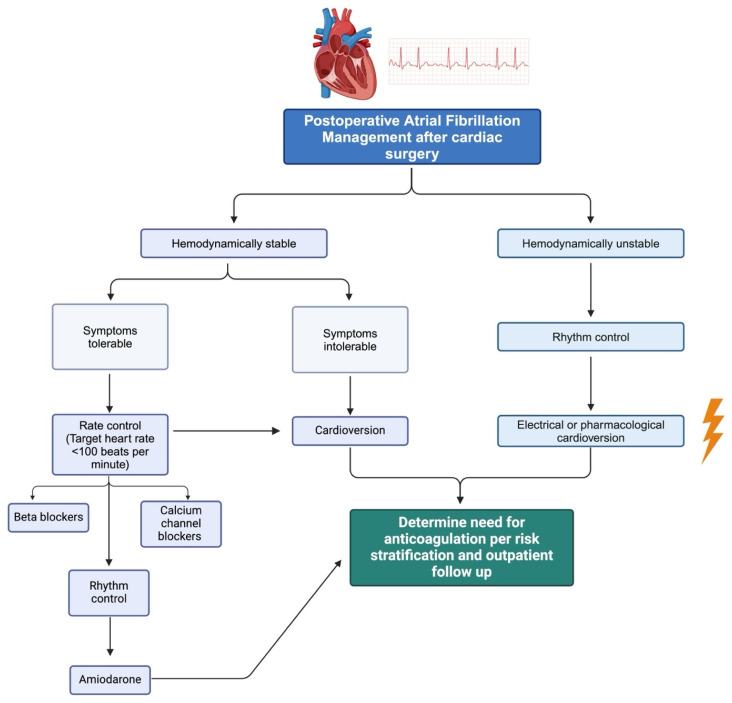
Algorithmic approach to POAF management. The process begins with the identification and diagnosis of POAF. Upon diagnosis, the initial step involves assessing the hemodynamic stability of the patient. If the patient is hemodynamically unstable, immediate synchronized cardioversion is indicated. For hemodynamically stable patients, the algorithm proceeds to evaluate the rate versus rhythm control strategy based on the duration and symptoms of POAF. Further steps include the consideration of anticoagulation therapy, determined by assessing thromboembolic risk and further outpatient management.

**Figure 5 biomedicines-12-01968-f005:**
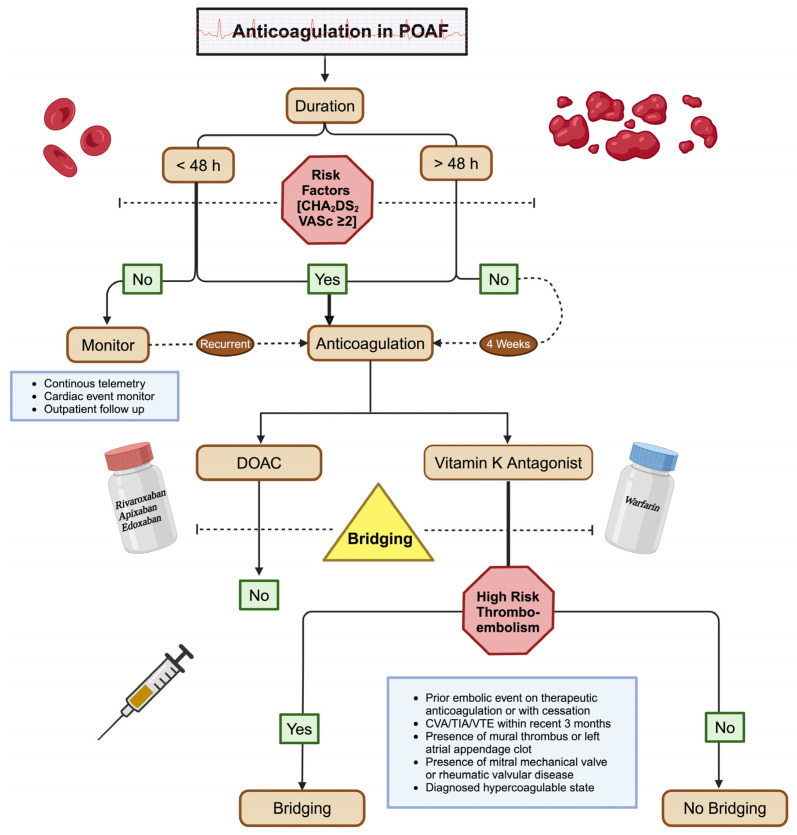
Algorithm for anticoagulation and bridging in POAF. The type of anticoagulation is determined based on patient preference, bleeding risks, and indications. Historically, warfarin is used for bridging in high-risk thromboembolism groups per American College of Clinical Pharmacy/American Heart Association recommendations. Note that if these high-risk factors are not present then bridging is advised against as there is an increased incidence of bleeding and adverse cardiovascular events with no significant benefit.

**Table 1 biomedicines-12-01968-t001:** Use of AI for POAF prediction.

Author	Aim of Study	Outcome
Karri et al. [46]	Compared the performance of a ML model with the established gold standard POAF Score in predicting POAF following cardiac surgery.	A study of 6040 patients found POAF in 21.5% (1364 admissions). ML models demonstrated superior predictive performance for POAF during ICU admission after cardiac surgery compared to the POAF Score, with AUCs as follows: GBM (0.74), LR (0.73), RF (0.72), KNN (0.68), SVM (0.67), and DT (0.59). The POAF Score achieved an AUC of 0.63.
Lu et al. [47]	Used ML algorithms to develop an efficient forecasting model for atrial fibrillation following cardiac surgery and compare the predictive performance of these algorithms with traditional logistic regression.	The study included 1400 patients who underwent valve and/or CABG with cardiopulmonary bypass. Postoperative atrial fibrillation occurred in 519 patients (37.1%). Predictive model AUCs were 0.777 (SVM), 0.767 (LR), and 0.765 (GBDT), with decision curve analysis showing appropriate net benefit for all models.
Magee et al. [48]	Developed an algorithm to predict the relative risk of developing postoperative atrial fibrillation in patients undergoing CABG.	Data from 19,083 patients undergoing CABG (1995–2006) were used to develop a logistic regression model with 14 significant indicators, including age, prolonged ventilation, cardiopulmonary bypass, and preoperative arrhythmias. The model showed 72.3% concordance, an AUC of 0.72, and a Hosmer–Lemeshow probability of 0.19. Calculated AF risk was 0.179 ± 0.116 for non-AF patients and 0.284 ± 0.153 for AF patients (*p* < 0.001).
He et al. [49]	Collected long-term single-lead ECGs of patients with preoperative sinus rhythm to develop statistical and ML models for predicting POAF.	The study of 100 cardiac surgery patients found POAF detection rates of 31% with long-term ECG and 19% with conventional monitoring. Significant differences in P-wave parameters were noted. The clinical model had an AUC of 0.86, while the clinical + ECG model had an AUC of 0.89. The SVM model achieved over 80% accuracy in the training set and over 60% in the test set.
Hiraoka et al. [50]	Develop an algorithm for immediate AF detection using an Apple Watch with a PPG sensor in cardiac surgery patients. ML is applied to the pulse data from the device to diagnose AF.	A total of 79 cardiac surgery patients were analyzed for POAF using telemetry-monitored ECGs and an Apple Watch. AF developed in 27 patients (34.2%), with 199 total AF events observed. The ML diagnostic algorithm on Apple Watch pulse data achieved an accuracy of 0.9416, with a sensitivity of 0.909 and specificity of 0.838.
Parise et al. [51]	Developed a ML prediction model of new-onset POAF following CABG.	This retrospective study of 394 patients undergoing first-time CABG developed an RF model to predict POAF, identifying key predictors: age (100%), preoperative creatinine (86.1%), aortic cross-clamping time (82.2%), body surface area (80.9%), Euro-Score (80.7%), and extracorporeal circulation time (65.7%). The RF model achieved the highest AUC values (up to 0.95), outperforming traditional logistic regression.
Tohyama et al. [52]	Developed a DL model using preoperative ECGs to predict POAF in patients undergoing surgery	This retrospective study analyzed 43,980 preoperative ECGs from 27,564 patients without AF. The model achieved a time-dependent C-statistic of 0.83 at 7 days, with 79.9% sensitivity, 73.5% specificity, and a 99.0% negative predictive value. The saliency map highlighted the importance of low-voltage P wave and ST regions, particularly in leads aVF, V1, V2, V5, and V6.
Oh et al. [53]	Developed a predictive model for POAF in non-cardiac surgery using ML.	The study used data from a cohort of 295,363. Key variables influencing POAF included age, lung operation, operation duration, history of coronary artery disease, and hypertension. The model achieved an AUC of 0.80, with 0.95 accuracy, 0.97 specificity, and 0.28 sensitivity.
Gruwez et al. [54]	Evaluated the usability of an AI-enabled ECG algorithm, originally trained to predict atrial fibrillation in non-surgical conditions, for predicting POAF in patients undergoing cardiac surgery.	The study analyzed 127 patients from the SURGICAL-AF trial who had no prior history of atrial fibrillation and had pre-operative 12-lead ECGs. The AI-enabled ECG algorithm predicted POAF with an AUC of 0.66, sensitivity of 64.3%, specificity of 64.7%, and accuracy of 0.65. POAF occurred in 40.4% of the high-risk group versus 21.4% of the low-risk group, indicating a hazard ratio of 2.2 (*p*-value = 0.020).
Rublev et al. [55]	Aimed to develop and compare ML models, particularly artificial neural networks and LR, for predicting POAF after on-pump CABG.	The study analyzed 866 patients who underwent isolated on-pump CABG surgery, excluding 85 with prior atrial fibrillation. POAF developed in 19.1% of cases. The best predictive model, an artificial neural network, identified 11 key risk factors and achieved an AUC of 0.75, specificity of 0.73, sensitivity of 0.74, and accuracy of 0.73.
Chamberlain [56]	Assessed whether the CHARGE-AF clinical risk score and an AI-ECG model can classify the risk of subsequent atrial fibrillation in patients with POAF after noncardiac surgery and to determine if a combined approach of both models improves risk prediction.	The study included 308 patients with POAF after noncardiac surgery. Subsequent AF rates were 87.16 and 198.51 per 1000 person-years for the lowest and highest tertiles of CHARGE-AF scores, respectively, and 90.93 and 226.00 per 1000 person-years for AI-ECG scores. The combined model had a C-statistic of 0.61, slightly improving prediction accuracy.
Zhang et al. [57]	Developed a robust AI-based tool for detecting AF and assessing AF burden using both surface ECG recordings and atrial electrograms in postoperative cardiac patients.	The study population consisted of 659 adult postoperative cardiac surgery patients, with data divided into training (263 patients), validation (66 patients), and testing (330 patients) sets. The AI tool achieved an AUC of 0.932 on validation and 0.953 on testing, with testing sensitivity of 97%, specificity of 81.4%, and an intraclass correlation coefficient of 0.952 for AF burden detection.
Siontis et al. [58]	Aimed to assess the performance of an AI-ECG algorithm in predicting POAF in patients undergoing noncardiac surgery and CABG, compared to its previous performance in a general population.	The study population included 342 patients with POAF and 255 controls from noncardiac surgery, and 4561 patients undergoing CABG, of whom 1437 had POAF. The AI-ECG model achieved a sensitivity of 75% and specificity of 49% for noncardiac surgery POAF (AUC 0.66), and a sensitivity of 50% and specificity of 61% for coronary surgery POAF (AUC 0.58), demonstrating lower performance compared to its original setting.

Random forest classifier (RF), Gradient-boosted machine (GBM), Logistic regression (LR), K-neighbors classifier (KNN), Support vector machine (SVM), Decision tree classifier (DT), Gradient-boosting decision tree (GBDT), and Photoplethysmography (PPG).

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
