# Peer review of "Postoperative Atrial Fibrillation: A Review"

_biomedicines, 2024, doi:10.3390/biomedicines12091968_

Round 1
Reviewer 1 Report
Comments and Suggestions for Authors
Authoritative review paper on postoperative AF, encompassing all the relevant aspects of the topic with up to date literature evidence and stimulating future research directions. All in all worth of publication in the present form.
Author Response
Comment 1: Authoritative review paper on postoperative AF, encompassing all the relevant aspects of the topic with up to date literature evidence and stimulating future research directions. All in all worth of publication in the present form.
Response 1: Thank you so much for reviewing our paper! We appreciate the wonderful feedback.
Reviewer 2 Report
Comments and Suggestions for Authors
The article by Shah et al. provides a comprehensive review of postoperative atrial fibrillation (POAF), a common and significant complication after surgery, particularly in cardiac procedures. The authors discuss the incidence, pathophysiology, risk factors, and management strategies related to POAF. Although the abstract mentions controversies around anticoagulation in short-lived POAF, the article could benefit from a more in-depth discussion of these areas. For instance, the decision-making process regarding anticoagulation in patients with varying bleeding risks could be explored further, perhaps by comparing guidelines or presenting expert opinions.
The article has a tendency to make broad generalizations about findings across different types of procedures, without adequately acknowledging the specific complexities that may arise in diverse surgical circumstances. For instance, there may be considerable differences in the etiology and therapy of POAF between cardiac and non-cardiac procedures, and it would be beneficial to clearly express this differential.
Author Response
Please see the attachment uploaded.

Round 2
Reviewer 2 Report
Comments and Suggestions for Authors
The authors answered all my questions